# BLOCK-DIAGONAL HESSIAN-FREE OPTIMIZATION FOR TRAINING NEURAL NETWORKS

## ABSTRACT

Second-order methods for neural network optimization have several advantages over methods based on first-order gradient descent, including better scaling to large mini-batch sizes and fewer updates needed for convergence. But they are rarely applied to deep learning in practice because of high computational cost and the need for model-dependent algorithmic variations. We introduce a variant of the Hessian-free method that leverages a block-diagonal approximation of the generalized Gauss-Newton matrix. Our method computes the curvature approximation matrix only for pairs of parameters from the same layer or block of the neural network and performs conjugate gradient updates independently for each block. Experiments on deep autoencoders, deep convolutional networks, and multilayer LSTMs demonstrate better convergence and generalization compared to the original Hessian-free approach and the Adam method.

## 1 INTRODUCTION

Deep neural networks have shown great success in computer vision (He et al., 2016a) and natural language processing tasks (Hochreiter and Schmidhuber, 1997). These models are typically trained using first-order optimization methods like stochastic gradient descent (SGD) and its variants. The vanilla SGD does not incorporate any curvature information about the objective function, resulting in slow convergence in certain cases. Momentum (Qian, 1999; Nesterov, 2013; Sutskever et al., 2013) or adaptive gradient-based methods (Duchi et al., 2011; Kingma and Ba, 2014) are sometimes used to rectify these issues. These adaptive methods can be seen as implicitly computing finite-difference approximations to the diagonal entries of the Hessian matrix (LeCun et al., 1998).

A drawback of first-order methods in general, including adaptive ones, is that they perform best with small mini-batches (Dean et al., 2012; Zhang et al., 2015; Das et al., 2016; Recht et al., 2011; Chen et al., 2016). This limits available parallelism and makes distributed training difficult. Moreover, in the distributed setting the gradients must be accumulated after each training update and the communication among workers may become a major bottleneck. The optimization methods that scale best to the distributed setting are those that can reach convergence with few parameter updates. The weakness of the first-order methods on this metric extends even to the convex case, where it was shown to be the result of correlation between the gradients at different data points in a mini-batch leading to "overshooting" in the direction of the correlation (Takác et al., 2013).

In the case of deep neural networks, large mini-batch sizes lead to substantially increased generalization error (Keskar et al., 2016; Dinh et al., 2017). Although Goyal et al. (2016) recently successfully trained deep ResNets on the ImageNet dataset in one hour with mini-batch size as large as 8192 by using the momentum-SGD equipped with some well-designed hyper-parameters, it also showed there is severe performance decay for even larger mini-batch sizes, which indeed corroborates the difficulty of training with large mini-batches. These difficulties motivate us to revisit the second-order optimization methods, which use the Hessian or other curvature matrices to rectify the gradient direction. The second-order methods employ more information about the local structure of the loss function, as they approximate it quadratically rather than linearly, and they scale better to large mini-batch sizes.

However, finding the exact minimum of a quadratic approximation to a loss function is infeasible in most deep neural networks because it involves inverting an $N$-by-$N$ curvature matrix for a parameter count of $N$. The Hessian-free (HF) methods (Martens, 2010; Martens and Sutskever, 2011; Byrd et al.,

2011; 2012) minimize the quadratic function that locally approximates the loss using the conjugate gradient (CG) method instead. This involves evaluating a sequence of curvature-vector products rather than explicitly inverting—or even computing—the curvature matrix or Hessian. The Hessian-vector product can be calculated efficiently using one forward pass and one backward pass (Pearlmutter, 1994), while other curvature-vector products have similarly efficient algorithms (Schraudolph, 2002; Martens and Sutskever, 2012).

Normally, the HF method requires many hundreds of CG iterations for one update, which makes even a single optimization step fairly computationally expensive. Thus, when comparing HF to first-order methods, the benefit in terms of fewer iterations from incorporating curvature information often does not compensate for the added computational burden.

We propose using a block-diagonal approximation to the curvature matrix to improve Hessian-free convergence properties, inspired by several results that link these two concepts for other optimization methods. Collobert (2004) argues that when training a multilayer perceptron (MLP) with one hidden layer the gradient descent converges faster with the cross-entropy loss than with mean squared error because its Hessian is more closely block-diagonal. A block-diagonal approximation of the Fisher information matrix, one kind of curvature matrix, has also been shown to improve the performance of the online natural gradient method (Le Roux et al., 2008) for training a one-layer MLP.

The advantage of a block-diagonal Hessian-free method is that updates to certain subsets of the parameters are independent of the gradients for other subsets. This makes the subproblem separable and reduces the complexity of the local search space (Collobert, 2004). We hypothesize that using the block-diagonal approximation of the curvature matrix may make the Hessian-free method more robust to noise that results from using a relatively small mini-batch for curvature estimation.

In the cases of Collobert (2004) and Le Roux et al. (2008), the parameter blocks for which the Hessian or Fisher matrix is block-diagonal are composed of all weights and biases involved in computing the activation of each neuron in the hidden and output layers. Thus it equates to the statement that gradient interactions among weights that affect a single output neuron are greater than those between weights that affect two different neurons.

In order to strike a balance between the curvature information provided by additional Hessian terms and the potential benefits of a more nearly block-diagonal curvature matrix, and adapt the concept to more complex contemporary neural network models, we choose to treat each layer or submodule of a deep neural network as a parameter block instead. Thus, different from Collobert (2004) and Le Roux et al. (2008), our hypothesis then becomes that gradient interactions among weights in a single layer are more useful for training than those between weights in different layers.

We now introduce our block-diagonal Hessian-free method in detail, then test this hypothesis by comparing the performance of our method on a deep autoencoder, a deep convolutional network, and a multilayer LSTM to the original Hessian-free method (Martens, 2016) and the Adam method (Kingma and Ba, 2014).

## 2 THE BLOCK-DIAGONAL HESSIAN-FREE METHOD

In this section, we describe the block-diagonal HF method in detail and compare it with the original HF method (Martens, 2010; Martens and Sutskever, 2011).

Throughout the paper, we use boldface lowercase letters to denote column vectors, boldface capital letters to denote matrices or tensors, and the superscript $\top$ to denote the transpose. We denote an input sample and its label as $(\mathbf{x}, \mathbf{y})$, the output of the network as $\mathbf{f}(\mathbf{x}, \mathbf{w})$, and the loss as $\ell(\mathbf{y}, \mathbf{f}(\mathbf{x}, \mathbf{w}))$, where $\mathbf{w}$ refers to the network parameters flattened to a single vector.

### 2.1 THE BLOCK-DIAGONAL HESSIAN-FREE METHOD

We first recall how second-order optimization works. For each parameter update, the second-order method finds $\Delta\mathbf{w}$ that minimizes a local quadratic approximation $q(\mathbf{w} + \Delta\mathbf{w})$ of the objective function $\ell(\cdot)$ at point $\mathbf{w}$:

$$q(\mathbf{w} + \Delta\mathbf{w}) := \ell(\mathbf{w}) + \Delta\mathbf{w}^\top \nabla\ell(\mathbf{w}) + \frac{1}{2}\Delta\mathbf{w}^\top \mathbf{G}(\mathbf{w})\Delta\mathbf{w}, \tag{1}$$

where $\mathbf{G}(\mathbf{w})$ is some curvature matrix of $\ell(\cdot)$ at $\mathbf{w}$, such as the Hessian matrix or the generalized Gauss-Newton matrix (Martens and Sutskever, 2012). The resulting sub-problem of

$$\arg\min_{\Delta\mathbf{w}} \Delta\mathbf{w}^{\top}\nabla\ell + \frac{1}{2}\Delta\mathbf{w}^{\top}\mathbf{G}\Delta\mathbf{w} \tag{2}$$

is solved using conjugate gradient (CG), a procedure that only requires evaluating a series of matrix-vector products $\mathbf{G}\mathbf{v}$.

There exist efficient algorithms for computing these matrix-vector products given a computation-graph representation of the loss function. If the curvature matrix $\mathbf{G}$ is the Hessian matrix, (1) is the second-order Taylor expansion and the Hessian-vector product can be computed as the gradient of the directional derivative of the loss function in the direction of $\mathbf{v}$, operations also known as the L- and R-operators $\mathcal{L}\{\cdot\}$ and $\mathcal{R}_{\mathbf{v}}\{\cdot\}$ respectively:

$$\mathbf{H}\mathbf{v} = \frac{\partial^2\ell}{\partial^2\mathbf{w}}\mathbf{v} = \nabla_{\mathbf{w}}(\mathbf{v}^{\top}\nabla_{\mathbf{w}}\ell) = \mathcal{L}\{\mathcal{R}_{\mathbf{v}}\{\ell(\mathbf{w})\}\}. \tag{3}$$

The R-operator can be implemented as a single forward traversal of the computation graph (applying forward-mode automatic differentiation), while the L-operator requires a backward traversal (reverse-mode automatic differentiation) (Pearlmutter, 1994; Baydin et al., 2015). The Hessian-vector product can also be computed as the gradient of the dot product of a vector and the gradient; that method does not require the R-operator but has twice the computational cost.

However, the objective of deep neural networks is non-convex and the Hessian matrix may have a mixture of positive and negative eigenvalues, which makes the optimization problem (2) unstable. It is common to use the generalized Gauss-Newton matrix (Schraudolph, 2002) as a substitute curvature matrix, as it is always positive semidefinite if the objective function can be expressed as the composition of two functions $\ell(\mathbf{f}(\mathbf{w}))$ with $\ell$ convex, a property satisfied by most training objectives. For a curvature mini-batch of data $\mathcal{S}_c$, the generalized Gauss-Newton matrix is defined as

$$\mathbf{G} := \frac{1}{|\mathcal{S}_c|}\sum_{(\mathbf{x},\mathbf{y})\in\mathcal{S}_c}\mathbf{J}^{\top}\mathbf{H}_{\ell}\mathbf{J}, \tag{4}$$

where $\mathbf{J}$ is the Jacobian matrix of derivatives of network outputs with respect to the parameters $\mathbf{J} := \frac{\partial\mathbf{f}}{\partial\mathbf{w}}$ and $\mathbf{H}_{\ell}$ is the Hessian matrix of the objective with respect to the network outputs $\mathbf{H}_{\ell} := \frac{\partial^2\ell}{\partial^2\mathbf{f}}$. It is an approximation to the Hessian that results from dropping terms that involve second derivatives of $\mathbf{f}$ (Martens and Sutskever, 2012).

The Gauss-Newton vector product $\mathbf{G}\mathbf{v}$ can also be evaluated as

$$\mathbf{G}\mathbf{v} = (\mathbf{J}^{\top}\mathbf{H}_{\ell}\mathbf{J})\mathbf{v} = \mathbf{J}^{\top}\left(\mathbf{H}_{\ell}\left(\mathbf{J}\mathbf{v}\right)\right). \tag{5}$$

In an automatic differentiation package like Theano (Al-Rfou et al., 2016), this requires one forward-mode and one reverse-mode traversal of the computation graphs of each of $\ell(\mathbf{f})$ and $\mathbf{f}(\mathbf{w})$.

However, it is still inefficient to solve problem (2) for a deep neural network with a large number of parameters, so we propose the block-diagonal Hessian-free method. We first split the network parameters into a set of parameter blocks. For instance, each block may contain the parameters from one layer or a group of adjacent layers. Then the sub-problems corresponding to each block are solved separately, while their solutions are concatenated together to produce a single update.

Specifically, if there are $B$ blocks in total, the parameter vector can be rewritten as $\mathbf{w} = [\mathbf{w}_{(1)}; \mathbf{w}_{(2)}; \ldots; \mathbf{w}_{(B)}]$. Similarly, we split the gradient into blocks as $\nabla\ell(\mathbf{w}) = [\nabla_{(1)}\ell; \nabla_{(2)}\ell; \ldots; \nabla_{(B)}\ell]$, where $\nabla_{(b)}\ell$ is the vector that contains the gradient only with respect to the parameters in block $b$. We further split the curvature matrix into $B \times B$ square blocks and let $\mathbf{G}_{(b)}$ be the $b$-th diagonal block of $\mathbf{G}$. Then we obtain separate sub-problems for each block as follows:

$$\arg\min_{\Delta\mathbf{w}_{(1)}} \Delta\mathbf{w}_{(1)}^{\top}\nabla_{(1)}\ell + \frac{1}{2}\Delta\mathbf{w}_{(1)}^{\top}\mathbf{G}_{(1)}\Delta\mathbf{w}_{(1)},$$

$$\arg\min_{\Delta\mathbf{w}_{(2)}} \Delta\mathbf{w}_{(2)}^{\top}\nabla_{(2)}\ell + \frac{1}{2}\Delta\mathbf{w}_{(2)}^{\top}\mathbf{G}_{(2)}\Delta\mathbf{w}_{(2)},$$

$$\ldots,$$

$$\arg\min_{\Delta\mathbf{w}_{(B)}} \Delta\mathbf{w}_{(B)}^{\top}\nabla_{(B)}\ell + \frac{1}{2}\Delta\mathbf{w}_{(B)}^{\top}\mathbf{G}_{(B)}\Delta\mathbf{w}_{(B)}.$$

---

**Algorithm 1** Block-Diagonal Hessian-Free Method

---

**Input**: Training data set $\mathcal{S}_T = \{(\mathbf{x}_i, \mathbf{y}_i), i = 1, \ldots, |\mathcal{S}_T|\}$; Neural network output function $\mathbf{z}_i = \mathbf{f}(\mathbf{x}_i, \mathbf{w})$ with parameters $\mathbf{w}$ and loss function $\ell(\mathbf{z}_i, \mathbf{y}_i)$; Hyper-parameters: maximum loops `max_loops`, maximum conjugate gradient iterations `max_cg_iters`, CG stop criterion `cg_stop_criterion`, learning rate $\alpha$

**Block partition**: Partition the network parameters into $B$ blocks, i.e., $\mathbf{w} = [\mathbf{w}_{(1)}; \mathbf{w}_{(2)}; \ldots; \mathbf{w}_{(B)}]$

**For** $k = 1, \ldots,$ **max_loops**:

1. Choose a gradient mini-batch $\mathcal{S}_g \subset \mathcal{S}_T$ to calculate the gradient $\mathbf{g} = [\mathbf{g}_{(1)}; \ldots; \mathbf{g}_{(B)}]$

2. Choose a curvature mini-batch $\mathcal{S}_c \subset \mathcal{S}_g$ to calculate the curvature-vector product

3. CG iterations:
   For $b = 1, \ldots, B$, solve $\arg\min_{\Delta\mathbf{w}_{(b)}} \Delta\mathbf{w}_{(b)}^\top \nabla_{(b)}\ell + \frac{1}{2}\Delta\mathbf{w}_{(b)}^\top \mathbf{G}_{(b)}\Delta\mathbf{w}_{(b)}$ by CG with `max_cg_iters` and `cg_stop_criterion`. These suboptimizations may be performed in parallel.

4. Aggregate $\Delta\mathbf{w} \leftarrow [\Delta\mathbf{w}_{(1)}; \ldots; \Delta\mathbf{w}_{(B)}]$ and update $\mathbf{w} \leftarrow \mathbf{w} + \alpha\Delta\mathbf{w}$

---

We solve these sub-problems separately by conjugate gradient and concatenate their solutions together. Hence $\Delta\mathbf{w} = [\Delta\mathbf{w}_{(1)}; \ldots; \Delta\mathbf{w}_{(B)}]$ will be our update (see Algorithm 1).

The $b$-th sub-problem of the block-diagonal HF method is equivalent to minimizing the overall objective (1) with constraint $\Delta\mathbf{w}_j = 0$ for $j \notin (b)$, since the second-order term of such a constrained objective is zero for all terms in $\mathbf{G}$ not in $\mathbf{G}_{(b)}$. This confirms that block-diagonal HF as described above is equivalent to ordinary HF with the curvature matrix replaced by a block-diagonal approximation that includes only terms involving pairs of parameters from the same block.

The problem (2) has been separated into independent sub-problems for each block, reducing the dimensionality of the search space that CG needs to consider. Although we have $B$ sub-problems to solve for one update, each sub-problem has smaller size and requires fewer CG iterations. Hence, the total compute needs are on par with those of the HF method with the same mini-batch sizes; if the independent sub-problems can be executed in parallel (e.g., on multiple nodes in a distributed system), there is potential for up to $B$-fold speed improvement. As we demonstrate below, block-diagonal Hessian-free achieves better performance than the HF method on deep autoencoders, multilayer LSTMs, and deep CNNs.

## 2.2 Implementation Details

We partition the network parameters into blocks based on the architecture of the network. When partitioning the network parameters, we try to define roughly equal sized blocks. This allows each sub-problem to make roughly similar progress with the same number of CG iterations. We seek to partition the network such that parameters whose gradients we expect to be strongly correlated are part of the same block. For example, in our experiment we split the autoencoder network into two blocks: one for the encoder and one for the decoder. For the multilayer LSTM, we treat each layer of recurrent cells as a block. And for the deep CNN, we divide the convolutional layers into three contiguous blocks.

When solving the problem (2), we use truncated conjugate gradient (Yuan, 2000). This means we terminate the CG iteration before finding the local minimum. There are two reasons to do this truncation. First, CG iterations are expensive and later iterations of CG provide diminishing improvements. More importantly, when we use mini-batches to evaluate the curvature-vector product, early termination of CG keeps the update from overfitting to the specific mini-batch.

One way to reduce the computational burden of the HF method is to use smaller mini-batch sizes to evaluate the curvature-vector product while still using a large mini-batch to evaluate the objective and the gradient (Byrd et al., 2011; 2012; Kiros, 2013). Martens (2010) similarly implements the HF method using the full dataset to evaluate the objective and the gradient, and mini-batches to calculate the curvature-vector products. This is possible because Newton-like methods are more tolerant to approximations of the Hessian than they are to that of the gradient (Byrd et al., 2011). In

our implementation, the curvature mini-batch is chosen to be a strict subset of the gradient mini-batch as shown in Algorithm 1.

However, small mini-batches inevitably make the curvature estimation deviate from the true curvature, reducing the convergence benefits of the HF method over first-order optimization (Martens and Sutskever, 2012). In practice it is not trivial to choose a mini-batch size that balances accurate estimation of curvature and the computational burden (Byrd et al., 2011). The key to making Hessian-free methods, including block-diagonal Hessian-free, converge well with small curvature mini-batches is to use short CG runs to tackle mini-batch overfitting.

Martens (2010) suggests using factored Tikhonov damping to make the HF method more stable. With damping, $\hat{\mathbf{G}} := \mathbf{G} + d\mathbf{I}$ is used as the curvature matrix to make the curvature "more" positive definite, where $d$ is the intensity of damping. We also incorporate damping in many of our experiments. For the sake of comparison, we use the same damping strength for the HF method and the block-diagonal HF method and choose a fixed value for each experiment.

Another suggestion made by (Martens, 2010) is to use a form of "momentum" to accelerate the HF method. Here, momentum means initializing the CG algorithm with the last CG solution scaled by some constant close to 1, rather than initializing it randomly or to the zero vector. This change often brings additional speedup with little extra computation. We apply a fixed momentum value of $0.95$ for all experiments.

We also adopt fixed hyper-parameter settings across the experiments, rather than an adaptive schedule. One reason is that the statistics that control the adaptive hyper-parameter scheduling can cost more than the gradient and curvature-vector product evaluation, which makes the HF method even slower. Furthermore, these tricks are not independent and it is often unclear how to adjust and fit them to every scenario. Our fixed hyperparameters work well in practice across the three different neural network architectures we investigated.

## 3 RELATED WORK

The Hessian matrix is indefinite for nonconvex objectives, which makes the second-order method unstable as the local quadratic approximation becomes unbounded from below. (Martens and Sutskever, 2012) advocates using the generalized Gauss-Newton matrix (Schraudolph, 2002) as the curvature matrix instead, which is guaranteed to be positive semi-definite. Another way to circumvent the indefiniteness of the Hessian is to use the Fisher information matrix as the curvature matrix; this approach has been widely studied under the name "natural gradient descent" (Amari and Nagaoka, 2007; Amari, 1998; Pascanu and Bengio, 2014; Le Roux et al., 2008). In some cases these two curvature matrices are exactly equivalent (Pascanu and Bengio, 2014; Martens, 2016). It has also been argued that the negative eigenvalues of the full Hessian are helpful for finding parameters with lower energy, e.g., in the saddle-free Newton method (Dauphin et al., 2014) and in an approach that mixes the Hessian and Gauss-Newton matrices (He et al., 2016b).

Recently, Martens and Grosse (2015); Grosse and Martens (2016), and Ba et al. (2017) propose the K-FAC method to approximate the natural gradient using a block-diagonal or block-tridiagonal approximation to the *inverse* of the Fisher information matrix, and demonstrate the advantages over first-order methods of a specialized version of this optimizer tailored to deep convolutional networks. In their work, the parameters are partitioned into blocks of similar size and structure to those used in our method.

## 4 EXPERIMENTS

We evaluate the performance of the block-diagonal HF method on three deep architectures: a deep autoencoder on the MNIST dataset, a 3-layer LSTM for downsampled sequential MNIST classification, and a deep CNN based on the ResNet architecture for CIFAR10 classification. For all three experiments, we first compare the performance of the block-diagonal HF method with that of Adam (Kingma and Ba, 2014) to demonstrate that block-diagonal Hessian-free is able to handle large batch size more efficiently. We then demonstrate the advantage of the block-diagonal method over ordinary Hessian-free by comparing their performance at various curvature mini-batch sizes.

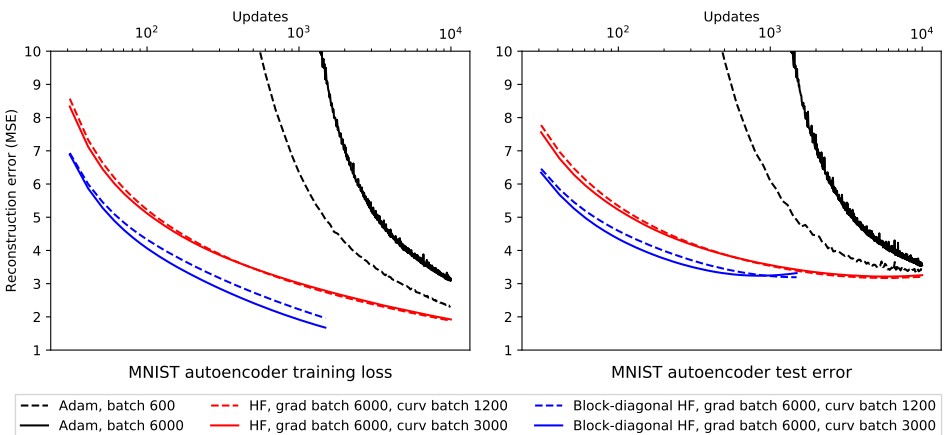

Figure 1: Performance comparison for a deep feedforward autoencoder on MNIST. The early epochs of Adam have reconstruction error greater than 10, while all models are run until the test error increases or for 1000 epochs.

Although the block-diagonal HF method needs to solve more quadratic minimization problems, each sub-problem is much smaller and the computation time is similar to the HF method. We note that the independence of the CG sub-problems means the block-diagonal method is particularly amenable to a distributed implementation.

We use the Lasagne deep learning framework (Dieleman et al., 2015) based on Theano (Al-Rfou et al., 2016) for our implementation of the HF and block-diagonal HF methods, as we found no other software framework to support both convenient definition of deep neural networks and the forward-mode automatic differentiation required to implement the R-operator.

## 4.1 Deep Autoencoder

Our first experiment is conducted on a deep autoencoder task. The goal of a neural network autoencoder is to learn a low-dimensional representation (encoding) of data from an input distribution. The "encoder" part, a multi-layer feedforward network, maps the input data to a low-dimensional vector representation while the "decoder" part, another multi-layer feedforward network, reconstructs the input data given the low-dimensional vector representation. The autoencoder is trained by minimizing the reconstruction error.

The MNIST dataset (LeCun et al., 2001) is composed of handwritten digits of size $28 \times 28$ with $60,000$ training samples and $10,000$ test samples. The pixel values of both the training and test data are rescaled to $[0, 1]$.

Our autoencoder is composed of an encoder with three hidden layers and state sizes 784-1000-500-250-30, followed by a decoder that is the mirror image of the encoder[1]. We use the tanh activation function and the mean squared error loss function.

For hyperparameters, we use a fixed learning rate of $0.1$, no damping, and maximum CG iterations max_cg_iters $= 30$ for both the HF and block-diagonal HF methods. For block-diagonal HF, we define two blocks: one block for the encoder and the other for the decoder. For Adam, we use the default setting in Lasagne with learning rate 0.001, $\beta_1$=0.9, $\beta_2$=0.999, and $\epsilon = 1 \times 10^{-8}$.

A performance comparison between Adam, HF, and block-diagonal HF is shown in Figure 1. For Adam, the number of dataset epochs needed to converge and the final achievable reconstruction error are heavily affected by the mini-batch size, with a similar number of updates required for small-mini-batch and large-mini-batch training. Our block-diagonal HF method with large mini-batch size achieves approximately the same reconstruction error as Adam with small mini-batches while requiring an order of magnitude fewer updates to converge compared to Adam with either small or large mini-batches. Moreover, block-diagonal Hessian-free provides consistently better reconstruction

---

[1] The model of autoencoder is the same as that in Hinton et al. (2006) and Martens (2010) for easy comparison.

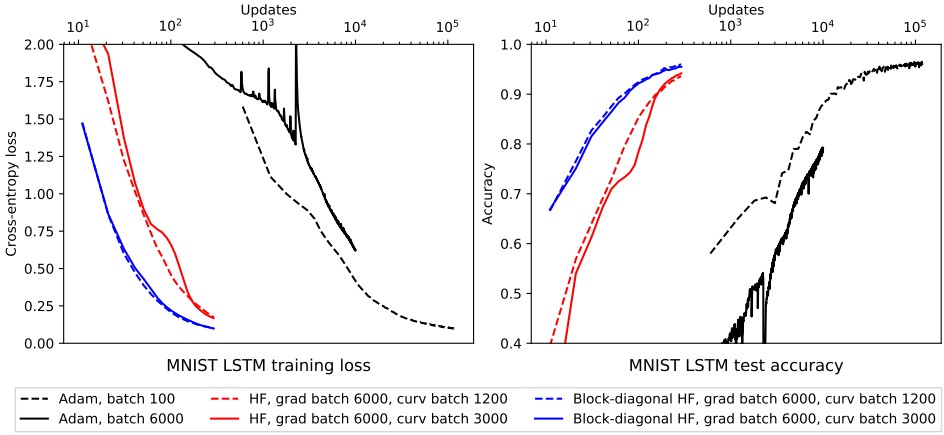

Figure 2: Performance comparison for a 3-layer stacked LSTM on the sequential MNIST $(7 \times 7)$ classification task. The early epochs of Adam have loss greater than 2, while all models are run until the test accuracy decreases or for 1000 epochs.

error—on both the train and test sets—than the HF method over the entire course of training. This advantage holds across different values of the curvature mini-batch size.

## 4.2 MULTILAYER LSTM

Our second experiment is conducted using a three-layer stacked LSTM on the sequential MNIST classification task. The MNIST data $(28 \times 28)$ is downsampled to $(7 \times 7)$ by average pooling. The neural network has three LSTM (Hochreiter and Schmidhuber, 1997; Gers et al., 2002) layers followed by a fully-connected layer on the final layer's last hidden state. Each LSTM has 10 hidden units with peephole connections (Gers et al., 2002).

For HF and block-diagonal HF, we use a fixed learning rate of $0.1$, damping strength $0.01$, and maximum CG iterations `max_cg_iter` $= 100$. The block-diagonal method has three blocks—one block for each LSTM layer, with the top block also containing the fully-connected layer. For Adam, we again use a learning rate of $0.001$, $\beta_1$=0.9, $\beta_2$=0.999, and $\epsilon = 1 \times 10^{-8}$.

A performance comparison between block-diagonal HF, HF, and Adam is found in Figure 2. Similar to the autoencoder case, the block-diagonal method with large mini-batches requires far fewer updates to achieve lower training loss and better test accuracy than Adam with any mini-batch size. Furthermore, compared to HF, the block-diagonal HF method requires fewer updates, achieves better minima, and exhibits less performance deterioration for small curvature mini-batch sizes.

## 4.3 DEEP CONVOLUTIONAL NEURAL NETWORK

We also train a deep convolutional neural network (CNN) for the CIFAR-10 classification task with the three optimization methods. The CIFAR-10 dataset has $50,000$ training samples and $10,000$ test samples, and each sample is a $32 \times 32$ image with three channels.

Our model is a simplified version of the ResNet architecture (He et al., 2016a). It has one convolutional layer $(16 \times 32 \times 32)$ at the bottom followed by three residual blocks and a fully-connected layer at the top. We did not include batch normalization layers[2].

For HF and block-diagonal HF, we use a fixed learning rate of $0.1$, damping strength $0.1$, and maximum CG iterations `max_cg_iter` $= 30$. The block-diagonal method again has three blocks—one for each residual block, with the top and bottom blocks also containing the fully-connected and convolution layers respectively. We use the same default Adam hyperparameters.

The common practice of training deep CNNs using custom-tuned learning rate decay schedules does not straightforwardly extend to the second-order case. However, Grosse and Martens (2016) suggests that Polyak averaging (Polyak and Juditsky, 1992) can obviate the need for learning rate decay while

---

[2] Computing the Hessian-vector product becomes extremely slow when involving the batch normalization layers with the Theano framework.

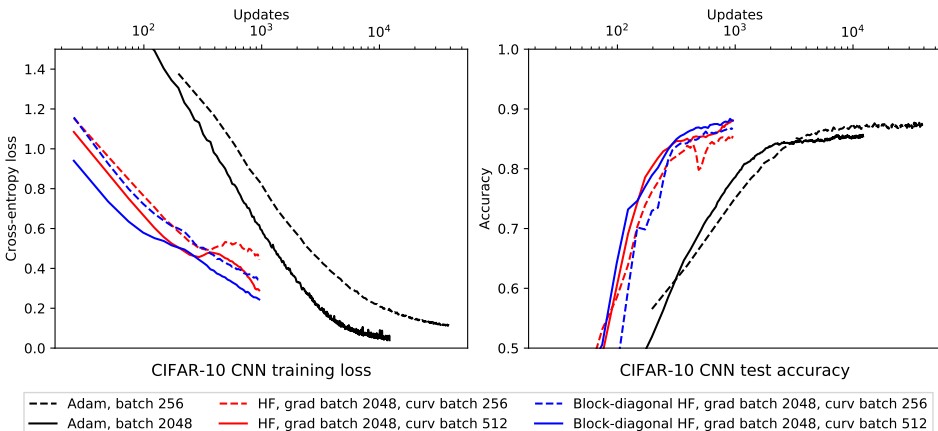

Figure 3: Performance comparison for a simplified residual CNN on the CIFAR-10 image classification task. The early epochs of Adam have loss greater than 1.5, while all models are run until the test accuracy decreases or for 1000 epochs.

still achieving high test accuracy. In order to ensure a fair comparison, we apply Polyak averaging with exponential decay rate 0.99 when evaluating the test accuracy for all three algorithms.

A performance comparison between block-diagonal HF, HF, and Adam is found in Figure 2. Similar to the autoencoder case, the block-diagonal method with large mini-batches requires far fewer updates to achieve lower training loss and better test accuracy than Adam with any mini-batch size. Furthermore, compared to the HF method, the block-diagonal HF method requires fewer updates, achieves better minima, and exhibits less performance deterioration for small curvature mini-batch sizes.

A performance comparison between block-diagonal Hessian-free, Hessian-free, and Adam is found in Figure 3. Block-diagonal HF with large mini-batches obtains comparable test accuracy to Adam with small ones. Furthermore, the block-diagonal method achieves slightly better training loss and higher test accuracy—and substantially more stable training—than Hessian-free for three different curvature mini-batch sizes.

Although not plotted in figures, the time consumption of block-diagonal HF and that of the HF are comparable in our experiments. The time per iteration of block-diagonal HF and HF is 5-10 times larger than that of the Adam method. However, the total number of iterations of block-diagonal HF and HF are much smaller than Adam and they have potential benefit of parallelization for large mini-batches.

## 5 Conclusion and Discussion

We propose a block-diagonal HF method for training neural networks. This approach divides network parameters into blocks, then separates the conjugate gradient subproblem independent for each parameter block. This extension to the original HF method reduces the number of updates needed for training several deep learning models while improving training stability and reaching better minima. Compared to first-order methods including the popular Adam optimizer, block-diagonal HF scales significantly better to large mini-batches, requiring an order of magnitude fewer updates in the large-batch regime.

Our results strengthen the claim of Collobert (2004) that "the more block-diagonal the Hessian, the easier it is to train" a neural network by showing that, in the case of Hessian-free optimization, simply ignoring off-block-diagonal curvature terms improves convergence properties.

Due to the separability of the subproblems for different parameter blocks, the block-diagonal HF method we introduce is inherently more parallelizable than the ordinary HF method. Future work can take advantage of this feature to apply the block-diagonal HF method to large-scale machine learning problems in a distributed setting.

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
