# OpenReview forum: "BLOCK-DIAGONAL HESSIAN-FREE OPTIMIZATION FOR TRAINING NEURAL NETWORKS"
_ICLR.cc/2018/Conference — Reject_

### Official Review · AnonReviewer3 · 2017-11-19
**There is nothing particularly wrong with the paper - it is a nice work, that is based a lot on previous attempts, like those in Martens. Coming from a more theoretical background, I would like to see more theory. Nevetheless this does not lessens the value of the paper. For the moment, weak accept until I also read the other reviews.**

**Rating:** 6
**Confidence:** 4

**Review:**

Summary:
The paper considers second-order optimization methods for training of neural networks.
In particular, the contribution of the paper is a Hessian-free method that works on blocks of parameters (this is a user defined splitting of the parameters in blocks, e.g., parameters of each layer is one block, or parameters in several layers could constitute a block).
This results into a block-diagonal approximation to the curvature matrix, in order to improve Hessian-free convergence properties: in the latter, a single step might require many CG steps, so the benefit from using second-order information is not apparent.
This is mainly an experimental work, where the authors show the merits of their approach on deep autoencoders, convolutional networks and LSTMs: results show favourable performance compared to the original Hessian-free approach and the Adam method.

Originality:
The paper is based on the works of Collobert (2004) and Le Roux et al. (2008), as well as the work of Martens: the twist is that each layer of the neural network is considered a parameter block, so that gradient interactions among weights in a single layer are more useful than those between weights in different layers. This increases the separability of the problem and reduces the complexity.

Importance:
Understanding the difference between first- and second-order methods for NN training is an important topic. Using second-order methods could be considered at its infancy, compared to the wide variety of first-order methods. Having new results on second-order methods with interesting results would definitely attract some attention at the conference.

Presentation/Clarity:
The paper is well structured and well written. The authors clearly place their work w.r.t. state of the art and previous works, so that it is clear what is new and what is known.

Comments:
1. It is not clear why the deficiency of first-order methods on training NNs with big batches motivates us to turn into second-order methods. Is there a reasoning for this statement? Or is it just because second-order methods are kind-of the only other alternative we have?

2. Assuming we can perform a second-order method, like Newton's method, on a deep NN. Since originally Newton's method was designed to find solutions that have gradient equal to zero, and since NNs have saddle points (probably many more than local minima), even if we could perfectly perform second-order Newton motions, there is no guarantee whether we converge to a local minimum or a saddle point. However, since we perform Newton's method approximately in practice, this might help escaping saddle points. Any comment on this aspect (I'm not aware whether this is already commented in Schraudolph 2002, where the Gauss-Newton matrix was proposed instead of the Hessian)?

---

> ### Author Response · Authors · 2018-01-03
> **RE: There is nothing particularly wrong with the paper - it is a nice work, that is based a lot on previous attempts, like those in Martens. Coming from a more theoretical background, I would like to see more theory. Nevetheless this does not lessens the value of the paper. For the moment, weak accept until I also read the other reviews.**
>
> We would like to thank the reviewer for appreciating our contribution in this paper.
>
> ------It is not clear why the deficiency of first-order methods on training NNs with big batches motivates us to turn into second-order methods. Is there a reasoning for this statement? Or is it just because second-order methods are kind-of the only other alternative we have?
>
> Response: It have been shown that second-order methods worked well with big batch sizes and in fact small batch size will make the convergence of the second-order methods unstable and hurt their performances. On the contrary, the first-order methods on training NNs with big batches have problem on the speedup and generalization (Keshar et al. 2016; Takac et al. 2013; Dinh et al. 2017). These  deficiencies of first-order methods with large mini batch size motivate us to turn into second-order methods to handle big batch size.
>
> ------Assuming we can perform a second-order method, like Newton's method, on a deep NN. Since originally Newton's method was designed to find solutions that have gradient equal to zero, and since NNs have saddle points (probably many more than local minima), even if we could perfectly perform second-order Newton motions, there is no guarantee whether we converge to a local minimum or a saddle point. However, since we perform Newton's method approximately in practice, this might help escaping saddle points. Any comment on this aspect (I'm not aware whether this is already commented in Schraudolph 2002, where the Gauss-Newton matrix was proposed instead of the Hessian)?
>
> Response: The reviewer proposes an very interesting view of the possible advantage of the Gauss-Newton matrix and the approximate Newton over Newton’s method, which was not commented in (Schraudolph 2002). As far as we know the main problem of Newton’s method on trading  deep NN is that for nonlinear system, the Hessian matrix is not necessarily positive definite so Newton’s method may diverge, which is consistent with the unstable practical performance of Newton’s method in training deep NN. The Gauss-Newton matrix is an approximation of the local curvature with positive semidefinite property as long as the loss function with respect to  the output of the network is convex, which holds for most popular loss functions (MSE, cross entropy). Indeed, these approximations to the curvature matrix may act as noises which help to escape saddle points while the exact Newton’s method may fail. This perspective  requires further exploration.

---

### Official Review · AnonReviewer1 · 2017-11-26
**The paper proposes a block-diagonal second order method for training deep networks. The algorithm is not  novel. Experimental results are good in training auto-encoder and LSTM (in terms of number of updates).**

**Rating:** 4
**Confidence:** 4

**Review:**

The paper proposes a block-diagonal hessian-free method for training deep networks.

- The block-diagonal approximation has been used in [1]. Although [1] is using Gauss-Newton matrix, the idea of "block-diagonal" approximation is similar.

- Is the computational time (per iteration) of the proposed method similar to SGD/Adam? All the figures are showing the comparison in terms of number of updates, but it is not clear whether this speedup can be reflected in the training time.

- Comparing block-diagonal approximation vs original HF method:
It is not clear to me what's the benefit using block-diagonal approximation. Is the time cost per iteration similar or faster?
Or the main benefit is to reduce #CG iterations? (but it seems #CG iterations are fixed for both methods in the experiments).
Also, the paper mentioned that "the HF method requires many hundreds of CG iterations for one update". Is this true?
 Usually we can set a stopping condition for solving the Newton system.

- It seems the benefit of block-diagonal approximation is marginal in CNN.

[1] Practical Gauss-Newton Optimisation for Deep Learning. ICML 2017.

---

> ### Author Response · Authors · 2018-01-03
> **RE: The paper proposes a block-diagonal second order method for training deep networks. The algorithm is not novel. Experimental results are good in training auto-encoder and LSTM (in terms of number of updates).**
>
> Thank the reviewer for the thoughtful feedback.
>
> ------The block-diagonal approximation has been used in [1]. Although [1] is using Gauss-Newton(GN) matrix, the idea of "block-diagonal" approximation is similar.
>
> Response: The work [1] and earlier works [2,3] uses block-diagonal approximation to the Gauss-Newton matrix or Fisher information matrix and requires explicit representation and inverse of each block of GN matrix or Fisher matrix. [1, 2] are only applied to feedforward networks. Further approximation and assumption are considered for working around convolutional neural networks [3]. It is not obvious how to generalize the algorithm to recurrent networks. The significant difference between ours and [1,2,3] is that we use block-diagonal approximation when evaluating the Hessian (Gauss-Newton) vector product and don’t require explicit inverse of the GN matrix and can work directly with convolutional and recurrent networks.
>
> ------Is the computational time (per iteration) of the proposed method similar to SGD/Adam? All the figures are showing the comparison in terms of number of updates, but it is not clear whether this speedup can be reflected in the training time.
>
> Response: The time consumption of block-diagonal Hessian-free (BHF) and that of the full Hessian-free (HF) are comparable. The time per iteration of BHF and HF is 5-10 times of the Adam method. However, the total number of iterations of BHF and HF are much smaller than Adam, which can offset the per-iteration cost.
>
>
> ------Comparing block-diagonal approximation vs original HF method: It is not clear to me what's the benefit using block-diagonal approximation. Is the time cost per iteration similar or faster? Or the main benefit is to reduce #CG iterations? (but it seems #CG iterations are fixed for both methods in the experiments). Also, the paper mentioned that "the HF method requires many hundreds of CG iterations for one update". Is this true? Usually we can set a stopping condition for solving the Newton system.
>
> Response: The BHF algorithm partitions the original HF method into a bunch of sub-problems and solves each sub-problem with the same CG iterations as the full HF method and hence may get more accurate solution. In practice given a fixed budgets of CGs the BHF takes slightly more time (15%) per update than the full HF method if without paralleling the sub-problems onto different workers but achieves better accuracy. Hence the main benefit is to reduce #CGs and achieve better accuracy. We note that the performance of HF cannot be improved by simply increasing the #CGs. It is true that we can set a stopping condition (fixed number of CGs) for solving the Newton system. How to achieve good accuracy given a number of CGs for solving the Newton system is not clear. Our BHF algorithm provides a way that easily achieve good accuracy with a small number of CG runs.
>
>
>
> [1] Practical Gauss-Newton Optimisation for Deep Learning. ICML 2017.
> [2] Optimizing Neural Networks with Kronecker-factored Approximate Curvature. ICML 2015
> [3] A Kronecker-factored Approximate Fisher Matrix for Convolution Layers. ICML 2016

---

### Official Review · AnonReviewer2 · 2017-11-27
**Nice paper, however, I would be happier if more experiments on larger datasets are presented**

**Rating:** 6
**Confidence:** 4

**Review:**

In this paper, authors discuss the use of block-diagonal hessian when computing the updates. The block-diagonal hessian makes it easier to solve the "newton" directions, as the CG can be run only on smaller blocks (and hence less CG iterations are needed).

The paper is nicely written and all was clear to me. In general, I agree that having larger batch-size is the way to go, for very large datasets and a pure SGD type of methods are having problems to efficiently utilize large clusters.

The only negative thing I find in the paper is the lack of more numerical results. Indeed, the paper is clearly not a theoretical paper, is proposing a new algorithm, hence there should be evidence that it works. For example, I would like to see how the choice of hyper-parameters influences the speed of the algorithm. Was "CG" cost included in the "x"-axis? i.e. if we put "passes" over the data as x-axis, then 1 update \approx 30 CG + some more == 32 batch evaluation.
So please try to make the "x"-axis more fair.

---

> ### Author Response · Authors · 2018-01-03
> **RE: Nice paper, however, I would be happier if more experiments on larger datasets are presented**
>
> We thank the reviewer for appreciating our work.
> The “x”-axis represents the number of updates and CG cost is not included.
> The time consumption of block-diagonal Hessian-free (BHF) and that of the full Hessian-free (HF) are comparable. The time per iteration of BHF and HF is 5-10 times of the Adam method. However, the total number of iterations of BHF and HF are much smaller than Adam, which can offset the per-iteration cost.

---

### Public Comment · (anonymous) · 2018-01-10
**More detail study and experiments**

1. Wall clock time plots:
     I think it is very important that you include in the final paper the wall-clock time plots. Per iteration, improvement is not of that much interest from a practical perspective if there is significantly different time computation. In the optimization of any Machine Learning model, there are two things of interest - if the final result is significantly different and how long it takes to get there. In your reply to AnonReviwer2, you mentioned that it is between 5-10 times of ADAM. However, it would be nice to clear up if you have actually measured that or is this an eye-balling of the factor. Also is this on a CPU or on a GPU? The aim of such publication should be towards people adopting these methods and without this **very** relevant information it really does not tell us a lot.

2. Batch size and model size
   In your first 2 experiments, you either use very large batch sizes (6000 or 1000) or very tiny models (LSTM with 10 hidden units). These very unrepresentative of what modern Deep Learning workloads are. Nevertheless, this is not nessacarily an issue, as the method is new. However, failing to report what happens when you decrease the batch size (very standard batch size is ~ 100-200) or increase the model size in the LSTM example (the smallest LSTMs I've seen used in practical applications have at least a few hundred nodes)  is missfortunate. Especially, from some of the papers in the literature, we know that using too small batches for the curvature matrices (when you can not afford to store moving averages like in [1,2,3]) can lead to significantly detrimental effects, as the Monte-Carlo estimates of the Gauss-Newton and the Hessian become degenerate. I genuinely hope that the authors decide to include this comparison in the final version. The final experiment in the paper is indeed interesting. However, as we can see ADAM achieves lower training cost. Although they have same test accuracy, this is really not much meaningful, as it is highly unfair to compare optimizer based on their generalization performance, when these are methods built for optimizing the objective at hand.

3. Hyperparameter optimization
   In the comparison against ADAM, you use the default setting of the optimizer. Although that is used often in practice, it is again unfair, since given that you have presented only 3 experiments, it is highly likely that the hyperparameters selected for the BHF have been highly tuned to those specific datasets. In [3] they specifically state that they also do hyperparameter search for ADAM parameters, including the decay rate of the learning rate. Additionally, on some of the CNN in the literature is not uncommon to use Momentum rather than ADAM. Thus a more in-depth comparison against better fine-tuned first order methods would be quite desirable.

[1] Optimizing Neural Networks with Kronecker-factored Approximate Curvature. ICML 2015
[2] A Kronecker-factored Approximate Fisher Matrix for Convolution Layers. ICML 2016
[3] Practical Gauss-Newton Optimisation for Deep Learning. ICML 2017.

---

### Decision · Program_Chairs · 2018-01-29
**ICLR 2018 Conference Acceptance Decision**

**Decision:**

Reject

**Comment:**

Pros:
+ Clearly written paper.

Cons:
- Limited empirical evaluation: paper should compare to first-order methods with well-tuned hyperparameters, since the block Hessian-free hyperparameters likely were well tuned, and plots of convergence as a function of time need to be included.
- Somewhat limited novelty in that block-diagonal curvature approximations have been used before, though the application to Hessian-free optimization is new.

The reviewers liked the clear description of the proposed algorithm and well-structured paper, but after discussion were not prepared to accept it primarily because (1) they wanted to see algorithmic comparisons in terms of convergence vs. time in addition to the convergence vs. updates that were provided; (2) they wanted more assurance that the baseline first-order optimizers had been carefully tuned; and (3) they wanted results on larger scale tasks.